# Effects of Organic and Inorganic Fertilizers on Soil Nutrient Conditions in Rice Fields with Varying Soil Fertility

**Guozhu Ma** [1,2,*] (ID), **Shenghai Cheng** [1,2], **Wenli He** [1,2], **Yixuan Dong** [3], **Shaowu Qi** [2,4,*], **Naimei Tu** [3] **and Weixu Tao** [1,2]

1  College of Tropical Crops, Hainan University, No. 58 Renmin Avenue, Meilan District, Haikou 570228, China; 21220951310008@hainanu.edu.cn (S.C.); 21220951310024@hainanu.edu.cn (W.H.); 20095131210069@hainanu.edu.cn (W.T.)
2  National Technology Innovation Center for Saline Tolerant Rice, No. 736 Yuanda 2nd Road, Furong District, Changsha 410125, China
3  College of Agriculture, Hunan Agricultural University, No. 1 Nongda Road, Furong District, Changsha 410127, China
4  Hunan Academy of Agricultural Sciences, No. 892 Yuanda 2nd Road, Furong District, Changsha 410125, China
*  Correspondence: student@hhrrc.ac.cn (G.M.); qishaowu@hhrrc.ac.cn (S.Q.)

**Abstract:** The majority of crop-growing areas in China have low or medium fertility levels, which limits the yield of crops grown in those areas. Fertilizer application can improve soil quality, but the effects of such treatments vary depending on the base soil fertility. However, the specific differences associated with the application of different fertilizer types to soils of varying fertility levels have yet to be clearly delineated. Here, the influences of several fertilizer types on physical, chemical, and biological soil indicators were assessed in rice fields in the red soil area of Hunan Province with varying base fertility levels: Hehua (low fertility), Dahu (medium fertility), and Longfu (high fertility). Four treatments were applied to these fields: no fertilizer, standard fertilizer, 60% chemical fertilizer + 40% organic fertilizer, and 100% chemical fertilizer. Across the three sites and treatment groups, the largest increases in total nitrogen and phosphorus contents were in Hehua and Longfu, respectively. Soil organic matter content increased most significantly in Hehua. Application of any type of fertilizer increased the total and fast-acting nutrient content in the low-yielding fields, whereas organic fertilizers increased the nutrient content and soil biological indicators more than chemical fertilizer alone did; the effect of organic fertilizer application on the combined enzyme activity of the soil was also higher than that of chemical fertilizers alone. Overall, these experiments provide a theoretical basis and technical support for rational fertilizer application and improvement of Hunan's red soil quality based on the natural soil fertility levels.

**Keywords:** base fertility; soil nutrients; enzyme activity; fertilizer application patterns

## 1. Introduction

Soil fertility is a fundamental parameter that determines the reproductive growth capacity, yield, and nutritional value of crop plants. In China, low- and medium-yielding fields account for ~67% of the total arable land area [1]. Improving soil quality in such fields to increase grain crop yields is an effective method of increasing food security and promoting the strategy of promoting the Chinese strategic national initiative of agricultural land and technique development [2]. Soil productivity, fertilizer application techniques, and soil improvement technology vary among areas with different base soil fertility rates [3]. Developing standard protocols for reasonable fertilizer application and soil improvement technology based on local soil fertility is, therefore, of great significance in rice production [4].

Rice-growing soils are influenced by the anthropogenic management practices associated with rice-based cropping systems [5]. These management practices affect parameters including soil capacitance, which affects the water-holding capacity and solute migration of a soil [6]. Soils with low capacitance and high porosity are conducive to root growth, aboveground tissue growth and development, and biomass accumulation, all of which can somewhat improve yield [7]. Li et al. [8] concluded that medium-, high-, and ultra-high-yielding fields generally have lower capacity than low-yielding fields and that soil porosity is highest in ultra-high-yielding fields, lower in high-yielding fields, and lowest in low-yielding fields. Soil management practices can also affect soil aggregate abundance. The number of aggregates reflects the ability of soil to supply and store nutrients [9]. The rate of soil aggregate destruction is correlated with organic matter content (i.e., high organic matter content is associated with low rates of aggregate destruction) [10,11]. High-yield, fertile soils generally have high organic matter content and low aggregate destruction rates [12]. In general, the responses of medium- and low-fertility soils to nitrogen fertilizers are more pronounced, whereas high-fertility soils have weaker responses. This is primarily due to differences in the chemical stability of agglomerates, which arise from the combined effects of salt solution concentrations and fertility levels [13].

Nitrogen (N) is a key nutrient required for crop growth and development. Soluble N is generally higher in high- and medium-fertility soils than in low-fertility soils. Tao et al. [14] showed that high-yield soils have relatively high levels of organic matter and alkaline nitrogen. Soil organic matter can be increased to promote soil total nitrogen and alkaline nitrogen fixation [15]. Shengxian et al. [5] concluded that soil organic matter, total N, available phosphorus content, and nutrients are highest in high-yielding rice soils, lower in medium-yielding rice soils, and lowest in low-yielding rice soils. However, Rui [16] determined that the main differences in soil performance are associated with variations in total N, organic matter, and fast-acting potassium content but not soil pH or available phosphorus content. Rui concluded that the discrepancies between their results and those of Shengxian et al. occurred because the two studies used soils with different textures and physical properties. Overall, low-fertility soils tend to have significantly lower cation exchange capacity, organic matter contents, and clay particle contents than high-fertility soils [17].

Previous studies have addressed the biological properties of soils with different fertility levels. For example, soil microbial carbon (C) is an important indicator of soil microbiological properties [18]. As soil fertility levels increase, soil microbial C and N levels increase accordingly [19]. One of the most important indicators of soil fertility is enzyme activity, which reflects the level of biological activity and the capacity for nutrient transformation, transport, and metabolism [20]. Soil enzymes are secreted by microorganisms, living plants, and animals, and they are released in the decomposition of plant and animal residues [21]. In high-fertility soil, increased N application is associated with an initial decrease, then an increase in peroxidase activity, whereas the opposite pattern is observed in medium-fertility soil. The level of N supply can indirectly reflect the level of urease activity, which is highly correlated with alkaline N levels [22]. Compared to medium-N conditions, under high-N conditions, increases in alkaline N content are associated with consistent or decreased urease activity, indicating that N fertilizer application affects urease [23]. Ye Xie Feng et al. [24] showed that tilling green manure significantly increased the enzymatic activity and fertility level of soil, and the highest enzymatic activity and fertility level was achieved when the tilling volume was 22,500–30,000 kg/hm$^2$. In low-fertility soils, the number and activity of microorganisms in the soil sink significantly increased with organic–inorganic application, which had a significant effect on improving soil fertility [25]. Liu et al. [26] showed that the long-term combined application of chemical fertilizers and pig manure could improve phosphatase activity in the soil, and the application of organic fertilizers could improve the soil structure and fertility. Similarly, when the straw application rate was 11,250 kg/hm$^2$, the number of fungi, bacteria, and actinomycetes and the activities of invertase and cellulase in the soil increased the most, and when the straw application rate

was 7500 kg/hm$^2$, soil alkaline urease, phosphatase activity, and alkaline nitrogen content increased significantly [27].

At present, the responses of rice soils with different base fertility levels to varied fertilizer treatments are unknown. As discussed above, prior results suggest that the effects of fertilization are highly dependent on soil properties at the physical, chemical, and biological levels. Thus, appropriate measures for soil quality improvement will depend on local soil conditions. Here, we examined these phenomena by using rice fields in the red soil areas of Hunan with variations in soil fertility. After assessing the base fertility levels and physicochemical and biological properties of three fields, we tested four fertilization treatments to clarify their effects on soil quality. Overall, the results of this study provide valuable new information regarding the efficiency of fertilizer application in soils with varying fertility levels and lay the foundation for improved rice cultivation technologies in eastern Hunan.

## 2. Materials and Methods

### 2.1. Test Materials and Sites

The rice variety 'Shenliangyou 5814' was obtained from Hunan Yahwa Seed Co. No. 11, Yangao Road, Lugu Hi-Tech Development Zone, Changsha City, Hunan Province, China and used in all experiments in this study. The first trial site was located in Baitang Village, Longfu Town, Liuyang City, in the eastern hilly region of Hunan Province (28°25′38.2″ N, 113°24′26.2″ E). The second was located in Niu Shi Ling Village, Hehua Office (28°034823.1″ N, 113°41′08.8″ E), and the third was located in Shuxiang Village, Dahu Town (28°52′35.1″ N, 113°54′09.5″ E). Each location contained a long-term soil fertility monitoring site established in 2013. The soil sampling time for this test was 2018. The soil at each site was red loam, and the tillage system was mono-annual.

### 2.2. Experimental Design

Four fertilizer treatments were tested at each site: no fertilizer (T1); standard fertilizer, which comprised 95% chemical fertilizer and 5% pig manure (T2); 60% chemical fertilizer + 40% organic fertilizer (T3); 100% chemical fertilizer (T4) (Table 1). Each treatment plot was 24 m$^2$ (6 × 4 m), arranged with a fully randomized design, and there were three biological replicates per treatment. Each experimental plot was constructed with field ridges (20 cm wide and 30 cm high) and wrapped with plastic film to prevent fertilizer and water infiltration between the plots, and each plot was single-rowed and single-irrigated. Other field management practices were consistent with those of the local one-season rice cropping system, including weed, pest, and disease control. The water management of the whole reproductive period was based on shallow water transplanting, inch water rejuvenation, shallow water tillering, sufficient seedlings for sunning, inch water for spike, and wet and strong seeds. The tested inorganic fertilizers were urea (46% N), calcium superphosphate (12% $P_2O_5$), and potassium chloride (60% $K_2O$); the organic fertilizers were pig manure (0.6% N, 0.4% $P_2O_5$, and 0.44% $K_2O$), zoysia (0.4% N, 0.1% $P_2O_5$, and 0.3% $K_2O$), and rice straw (0.6% N, 0.3% $P_2O_5$, and 1.1% $K_2O$) (Table 1). Organic fertilizers and inorganic phosphorus fertilizer were applied as base fertilizers. The inorganic N and K fertilizers were applied in stages: 50% base fertilizer, 30% at the tillering stage, and 20% at the spike stage.

**Table 1.** Fertilizer treatments that were applied at each test site. All measurements are in kg/hm$^2$.

| Treatment Group Name | Treatment Group Description | Inorganic Fertilizer | | | Organic Fertilizer | | | Total | | |
|---|---|---|---|---|---|---|---|---|---|---|
| | | N | P$_2$O$_5$ | K$_2$O | Pig Manure | Zoysia | Rice Straw | N | P$_2$O$_5$ | K$_2$O |
| T1 | No fertilizer | - | - | - | - | - | - | - | - | - |
| T2 | Standard fertilizer | 171.0 | 69.0 | 113.4 | 1500 | - | - | 180 | 75 | 120 |
| T3 | 40% organic fertilizer | 108.0 | 51.6 | 42.6 | - | 12,600 | 3600 | 180 | 75 | 120 |
| T4 | 100% chemical fertilizer | 180.0 | 75.0 | 120.0 | - | - | - | 180 | 75 | 120 |

*2.3. Soil Property Measurements*

Soil samples were collected at a depth of 0 to 20 cm at the tiller bloom stages, pregnancy spike stages, tassel stages, waxing stages, and maturity stages of rice. A five-point sampling method was used to extract the whole soil layer at each collection timepoint. All samples were dried at room temperature and then passed through 20- and 60-mesh sieves. Soil capacity was determined by using the ring knife method [28]. Basic soil nutrient indicators (total N, total phosphorus, organic matter content, available phosphorus, alkaline decomposition N, and pH) were determined with conventional analytical methods, as described in the Soil Agrochemical Analysis Methods [29] (Table 2). Urease activity was determined with the indophenol blue colorimetric method [30] and expressed in mg of ammoniacal N (NH$_3$-N) per g of soil after 24 h of incubation at 37 °C (mg·g$^{-1}$·d$^{-1}$). Phosphatase activity was determined by using the colorimetric method with sodium phenyl phosphate and expressed in mg of p-aminophenol per g of soil after 1 h of incubation at 37 °C (mg·g$^{-1}$·h$^{-1}$) [30]. Sucrose activity was determined with the 3,5 dinitro salicylic acid method and expressed in mg of glucose per g of soil after 24 h incubation at 37 °C (mg·g$^{-1}$·d$^{-1}$) [30].

**Table 2.** Basic rice soil chemical properties.

| Sample Site | pH | Organic Matter (g/kg) | Total Nitrogen (g/kg) | Total Phosphorus (g/kg) | Total Potassium (g/kg) | Available Nitrogen (mg/kg) | Available Phosphorus (mg/kg) | Available Potassium (mg/kg) |
|---|---|---|---|---|---|---|---|---|
| Hehua | 6.45 | 11.55 | 0.75 | 0.60 | 13.55 | 84.82 | 4.12 | 68.21 |
| Dahu | 5.65 | 25.45 | 1.35 | 0.84 | 8.03 | 155.38 | 3.63 | 72.34 |
| Longfu | 5.92 | 28.21 | 1.82 | 0.35 | 8.60 | 150.04 | 12.85 | 60.31 |

*2.4. Combined Soil Fertility Values*

Soil fertility was calculated by using the Nemero index method. The first step was the calculation of the partition fertility factor $\textbf{\textit{IFI}}_i$:

$$\textbf{\textit{IFI}}_i = \begin{cases} X/X_a & X \leq X_a \\ 1 + (X - X_a)/(X_c - X_a) & X_a < X \leq X_c \\ 2 + (X - X_c)/(X_P - X_c) & X_{C^C} < X \leq X_P \\ 3 & X > X_P \end{cases}$$

where $\textbf{\textit{IFI}}_i$ is the fertility factor; $X$ is the measured value of a given property; $X_a$ and $X_P$ are the lower and upper grading criteria, respectively (Table 3); $X_c$ is between the lower and upper ends of the attribute value grading scale such that $X_a < X_c < X_P$.

**Table 3.** Grading standards for soil properties. The criteria were consistent with those set forth in the Second National Soil Census.

| Grade | Organic Matter (g/kg) | Nitrogen Alkali Digestion (mg/kg) | Fast-Acting Potassium (mg/kg) | Effective Phosphorus (mg/kg) |
|---|---|---|---|---|
| $X_a$ | 10 | 60 | 40 | 3 |
| $X_c$ | 20 | 120 | 100 | 10 |
| $X_P$ | 30 | 180 | 150 | 20 |

The second step was the calculation of the combined soil fertility by using the Nemero formula as follows:

$$\text{IFI} = \sqrt{\frac{\left(\text{IFI}_i \text{ Average}\right)^2 + \left(\text{Minimum}\right)^2}{2} \times \left(\frac{n-1}{n}\right)}$$

where IFI is the combined soil fertility; IFIi average and IFIi minimum are the mean and minimum fertility values for a given soil attribute, respectively; n is the number of evaluation indicators.

*2.5. Soil Enzyme Activity Composite Index*

A combined soil enzyme activity index, GMea, was used:

$$\text{GMea} = \sqrt[3]{\text{Inv} \times \text{Ure} \times \text{Acp}}$$

where Inv is the sucrase activity, Ure is the urease activity, and Acp is the acid phosphatase activity.

*2.6. Statistical Analysis*

Statistical analyses were conducted in SPSS v23. Differences between treatment groups were assessed with analysis of variance (ANOVA) and considered statistically significant at $p < 0.05$. Graphs were generated in Excel 2016.

**3. Results**

*3.1. Effects of Fertilizer Treatments on Rice Soil Physical Properties*

We first assessed the effects of fertilizer application on the physical soil properties at the three sample sites. In the four treatments, the application of chemical fertilizers and organic fertilizers could reduce soil bulk density and increase soil porosity (Table 4). There were no significant differences in soil bulk density or soil porosity between the treatment groups at any of the sites. However, there were site-specific differences between these parameters; the soil at Longfu was denser and had higher porosity than the soil from Dahu, which, in turn, had higher density and porosity than the soil from Hehua. Soil with a bulk density of <0.9 g/cm$^3$ was considered to be too loose, whereas soil with a capacity of 1.0–1.2 g/cm$^3$ was considered to have a suitable texture for crop growth [31]. Therefore, Hehua soils from all treatment groups were too loose; all Dahu samples were suitable, except for the T1 samples; all Longfu samples were suitable.

**Table 4.** Effects of fertilizer treatment on rice soil bulk density and porosity.

| Treatment Group | Soil Capacity (g/cm³) | | | Soil Porosity (%) | | |
|---|---|---|---|---|---|---|
| | Hehua | Dahu | Longfu | Hehua | Dahu | Longfu |
| T1 | 0.99 ± 0.090 a | 0.79 ± 0.075 a | 1.19 ± 0.069 a | 62.56 ± 0.090 a | 58.83 ± 0.075 a | 58.84 ± 0.069 a |
| T2 | 1.02 ± 0.044 a | 1.09 ± 0.121 a | 1.09 ± 0.045 a | 61.55 ± 0.044 a | 57.95 ± 0.121 a | 60.25 ± 0.045 a |
| T3 | 1.05 ± 0.080 a | 1.11 ± 0.069 a | 1.05 ± 0.053 a | 60.50 ± 0.080 a | 57.14 ± 0.069 a | 59.64 ± 0.053 a |
| T4 | 0.93 ± 0.065 a | 1.14 ± 0.088 a | 1.07 ± 0.066 a | 64.74 ± 0.065 a | 54.94 ± 0.088 a | 57.39 ± 0.066 a |

Lowercase letters within a column indicate statistically significant groups at $p < 0.05$ (analysis of variance (ANOVA)).

### 3.2. Effects of Fertilizer Treatments on Rice Soil Nutrients

We next analyzed the effects of each fertilizer treatment on the chemical properties of each site. Across all treatments, Longfu generally had the highest total N and organic matter contents, whereas Dahu had the highest total phosphorus content (Table 5). The differences in total N and organic matter content were not significant between Hehua and Longfu. In the Hehua samples, the total N and organic matter contents were 10.13–14.47% and 11.54–19.07% higher, respectively, in the T4 treatment compared to the other treatments, and the total phosphorus content was 12.66–30.88% higher in the T3 treatment than in the other treatments. In Dahu samples, the total N, total phosphorus, and organic matter contents were significantly higher in the T3 treatment than in the control. In the Longfu samples, the total N, total phosphorus, and organic matter contents were 10.38–10.99%, 11.54–38.10%, and 0.33–7.59% higher, respectively, in the T2 treatment than in the other treatments.

**Table 5.** Effects of fertilizer treatment on total nutrients in rice soil at the three sites.

| Sample Site | Treatment Group | Total Nitrogen (g/kg) | Total Phosphorus (g/kg) | Organic Matter (g/kg) |
|---|---|---|---|---|
| Hehua | T1 | 0.76 ± 0.085 a | 0.68 ± 0.005 c | 11.85 ± 1.075 a |
| | T2 | 0.79 ± 0.034 a | 0.79 ± 0.020 b | 12.35 ± 1.553 a |
| | T3 | 0.77 ± 0.046 a | 0.89 ± 0.050 a | 12.62 ± 2.775 a |
| | T4 | 0.87 ± 0.040 a | 0.75 ± 0.015 b | 14.11 ± 2.259 a |
| Dahu | T1 | 1.00 ± 0.097 b | 0.92 ± 0.005 c | 24.90 ± 3.030 b |
| | T2 | 1.36 ± 0.119 a | 0.95 ± 0.011 b | 27.03 ± 1.235 ab |
| | T3 | 1.37 ± 0.149 a | 0.98 ± 0.016 a | 28.96 ± 0.325 a |
| | T4 | 1.36 ± 0.048 a | 0.94 ± 0.007 b | 26.21 ± 0.729 ab |
| Longfu | T1 | 1.82 ± 0.048 a | 0.42 ± 0.005 d | 28.47 ± 0.620 a |
| | T2 | 2.02 ± 0.188 a | 0.58 ± 0.002 a | 30.63 ± 1.156 a |
| | T3 | 1.83 ± 0.511 a | 0.52 ± 0.005 b | 30.53 ± 1.590 a |
| | T4 | 1.83 ± 0.118 a | 0.46 ± 0.005 c | 30.37 ± 4.468 a |

Lowercase letters within a column indicate statistically significant groups at $p < 0.05$ (ANOVA).

The alkaline N content was highest in the Longfu samples and lowest in the Hehua samples, with those from Dahu falling in the middle (Table 6). The maximum alkaline N content in the Hehua samples occurred at the tiller bloom stage, then gradually decreased as the fertile period progressed. The available phosphorus content was highest in Dahu and lowest in the Longfu samples; Hehua and Dahu reached the maximum available phosphorus levels at the tiller bloom and pregnancy spike stages, respectively. At maturity, both the Longfu and Dahu samples had significantly higher alkaline N content in all treatment groups than the Hehua samples did (by 73.6–142.8 mg/kg and 45.47–64.63 mg/kg, respectively).

**Table 6.** Effects of fertilizer treatment on fast-acting nutrient content in rice soil.

| Sample Site | Nutrient Indicator (mg/kg) | Treatment Group | Tiller Bloom | Pregnancy Spike | Tassel | Waxing | Maturity |
|---|---|---|---|---|---|---|---|
| Hehua | Alkaline nitrogen content | T1 | 94.03 ± 0.404 c | 83.07 ± 0.808 c | 97.98 ± 3.453 b | 77.70 ± 0.926 b | 83.90 ± 3.245 b |
| | | T2 | 114.33 ± 1.070 b | 100.57 ± 8.697 a | 98.58 ± 4.041 b | 91.00 ± 8.231 a | 104.53 ± 4.351 a |
| | | T3 | 117.37 ± 0.404 a | 95.55 ± 1.750 ab | 106.87 ± 1.125 a | 86.30 ± 1.424 ab | 90.77 ± 1.762 ab |
| | | T4 | 116.43 ± 1.070 a | 90.53 ± 0.808 bc | 102.62 ± 2.671 ab | 82.48 ± 4.829 ab | 90.07 ± 5.300 ab |
| | Effective phosphorus | T1 | 2.33 ± 0.858 d | 6.49 ± 3.250 b | 7.95 ± 1.032 c | 7.68 ± 2.575 d | 5.87 ± 2.730 c |
| | | T2 | 4.97 ± 1.741 c | 9.47 ± 1.487 a | 12.17 ± 4.763 b | 12.04 ± 0.991 b | 12.18 ± 2.951 a |
| | | T3 | 7.97 ± 1.983 b | 9.73 ± 2.988 a | 14.09 ± 6.157 a | 13.86 ± 4.814 a | 11.81 ± 1.247 a |
| | | T4 | 8.52 ± 0.744 a | 9.10 ± 3.481 a | 15.25 ± 2.543 a | 10.42 ± 2.730 c | 9.37 ± 0.991 b |
| Dahu | Alkaline nitrogen content | T1 | 133.01 ± 7.000 c | 104.07 ± 8.640 c | 149.92 ± 14.534 c | 125.07 ± 4.554 c | 157.50 ± 2.425 c |
| | | T2 | 175.11 ± 7.000 b | 162.40 ± 1.852 b | 173.25 ± 3.654 b | 212.68 ± 4.057 a | 205.10 ± 0.700 b |
| | | T3 | 189.23 ± 0.001 b | 196.00 ± 3.051 a | 204.35 ± 13.376 a | 204.98 ± 3.909 a | 233.57 ± 8.015 a |
| | | T4 | 217.63 ± 13.301 a | 191.33 ± 2.650 a | 221.78 ± 2.627 a | 161.82 ± 1.654 b | 218.40 ± 4.850 ab |
| | Effective phosphorus | T1 | 1.44 ± 2.623 c | 2.66 ± 1.593 d | 4.18 ± 1.741 d | 2.86 ± 2.064 d | 3.22 ± 0.496 d |
| | | T2 | 3.39 ± 1.032 a | 11.71 ± 4.157 a | 9.66 ± 3.574 a | 12.01 ± 2.818 a | 13.56 ± 3.516 a |
| | | T3 | 4.21 ± 3.469 a | 9.50 ± 0.572 b | 7.52 ± 1.247 b | 7.45 ± 1.032 b | 8.08 ± 1.311 b |
| | | T4 | 2.33 ± 1.487 b | 5.70 ± 2.760 c | 4.64 ± 1.359 c | 4.11 ± 2.160 c | 5.63 ± 1.593 c |
| Longfu | Alkaline nitrogen content | T1 | 143.80 ± 2.211 c | 142.63 ± 1.779 b | 151.67 ± 9.812 b | 132.07 ± 5.658 a | 142.57 ± 6.274 b |
| | | T2 | 150.73 ± 3.523 b | 146.07 ± 10.200 b | 163.10 ± 2.145 ab | 143.97 ± 2.458 a | 150.97 ± 4.102 ab |
| | | T3 | 157.97 ± 5.064 a | 148.63 ± 0.809 ab | 187.017 ± 4.700 a | 145.83 ± 2.977 a | 155.40 ± 0.700 a |
| | | T4 | 146.17 ± 1.070 c | 157.03 ± 2.139 a | 151.73 ± 2.357 b | 148.17 ± 2.6501 a | 152.83 ± 6.870 a |
| | Effective phosphorus | T1 | 10.39 ± 1.741 b | 20.11 ± 8.058 b | 19.08 ± 2.271 c | 18.29 ± 7.027 b | 17.36 ± 3.300 c |
| | | T2 | 11.78 ± 4.965 b | 20.93 ± 0.572 b | 19.61 ± 4.989 bc | 18.78 ± 3.095 b | 18.78 ± 3.469 b |
| | | T3 | 14.32 ± 1.787 a | 25.33 ± 1.314 a | 22.16 ± 1.983 a | 17.36 ± 2.904 b | 21.66 ± 0.858 a |
| | | T4 | 11.32 ± 5.253 b | 20.64 ± 2.904 b | 20.24 ± 0.572 b | 21.85 ± 2.976 a | 18.42 ± 3.753 bc |

Lowercase letters within a column indicate statistically significant groups at $p < 0.05$ (ANOVA).

### 3.3. Effects of Fertilizer Treatment on Combined Soil Fertility Values

We next analyzed site-specific and treatment-induced differences in soil fertility. Consistently, the IFI values were higher for Longfu and Dahu than for the Hehua samples. Furthermore, as expected, the IFI values were higher in all fertilized treatment groups (T2–T4) than in the control (T1) at each site (Figure 1). Overall, the Longfu T2 samples had the highest IFI (1.85), followed by Dahu T3 (1.80). In the Hehua samples, the highest IFI value was in T4, followed by T2, then T3; Longfu had the highest IFI value in T2, followed by T3, then T4; Dahu had the highest IFI value in T3, followed by T4, then T2.

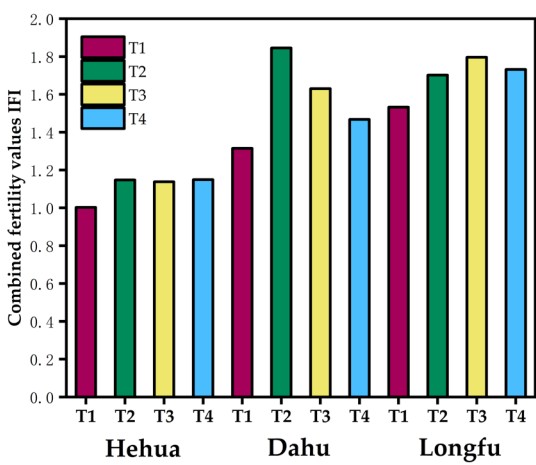

**Figure 1.** Combined soil fertility values in samples collected from the Hehua, Longfu, and Dahu sites. T1, no fertilizer; T2, standard fertilizer; T3, 40% organic fertilizer; T4, 100% chemical fertilizer.

### 3.4. Effects of Fertilizer Treatment on Enzyme Activity in Rice Soil

To establish the effects of different types of fertilizer treatment on rice soil enzymes, we quantified the activities of three key nutrient cycling enzymes: urease, acid phosphatase, and sucrase.

### 3.4.1. Urease

At the maturity stage, the urease activity was highest in Longfu and lowest in the Hehua samples overall (Figure 2). In the soil collected from Hehua, the T4 samples had the highest urease activity in the early stages of fertility; at the tiller bloom stage, the urease activity was 9.43%, 1.52%, and 1.80% higher in T4 compared to the T1–T3 samples, respectively, and at the pregnancy spike stage, the urease activity was 25.4%, 12.5%, and 1.82% higher in T4 than in the T1–T3 samples, respectively. At later growth stages, the T2 and T3 treatments had higher urease activity. Specifically, at the tassel and waxing stages, the highest urease activities were found in the T3 samples ($0.62 \text{ mg·g}^{-1}\text{·d}^{-1}$) and in the T2 samples ($0.72 \text{ mg·g}^{-1}\text{·d}^{-1}$), respectively. At the maturity stage, the T3 treatment had the highest activity at $0.57 \text{ mg·g}^{-1}\text{·d}^{-1}$; this was higher than in the T1, T2, and T4 samples by 31.18%, 10.26%, and 22.35%, respectively.

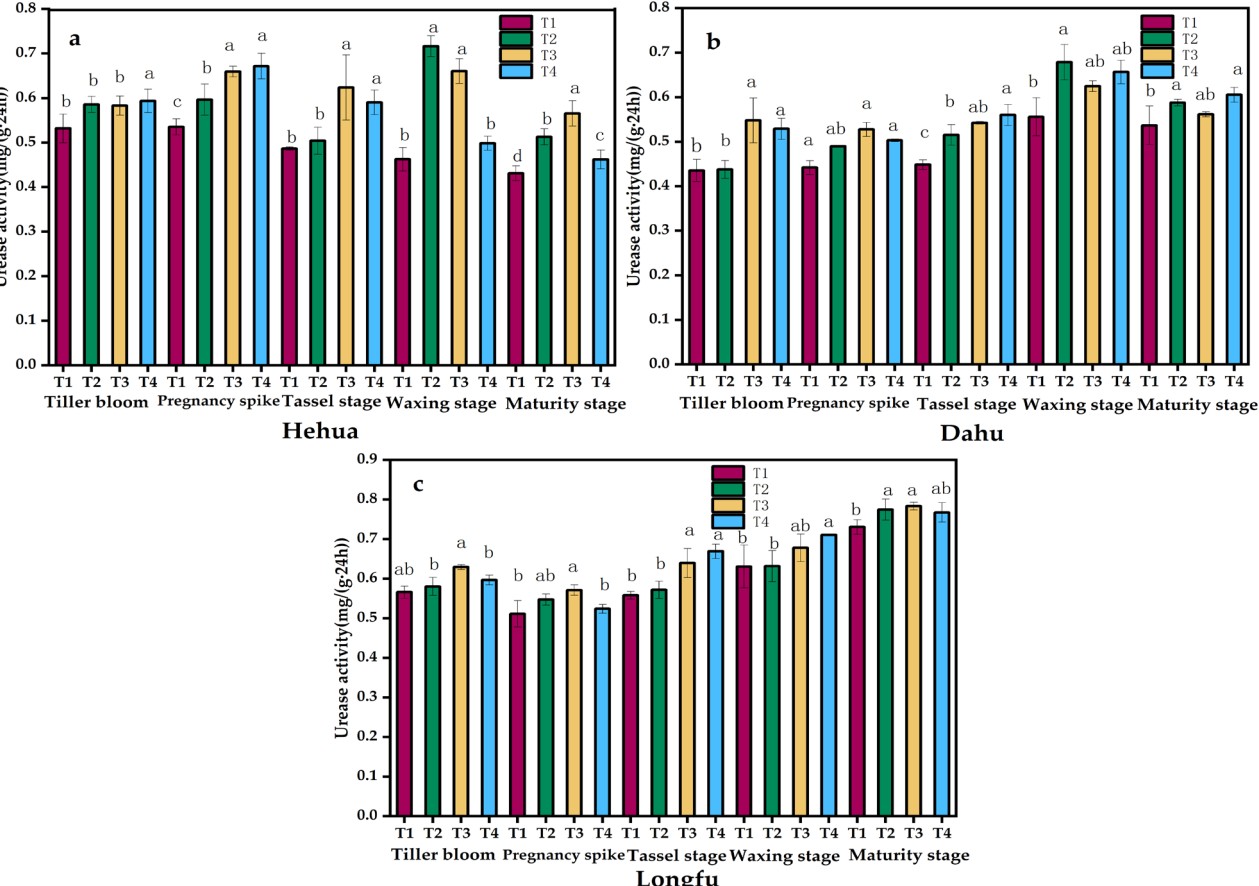

**Figure 2.** At each reproductive stage of rice development (**a**) is Urease activity in soil samples collected from Hehua; (**b**) is Urease activity in soil samples collected from Dahu; (**c**) is Urease activity in soil samples collected from Longfu. T1, no fertilizer; T2, standard fertilizer; T3, 40% organic fertilizer; T4, 100% chemical fertilizer. Lowercase letters above each bar indicate statistically significant groups at $p < 0.05$ (analysis of variance (ANOVA)).

In the soil collected from Dahu, the urease activity was highest in the T3 samples. At the tillering stage, the urease activity was 25.84% higher in the T3 than the T1 samples; at the pregnancy spike stage, it was 19.40% higher in the T3 samples than in the T1 samples. At the tassel stage, the urease activity was highest in the T4 treatment ($0.56 \text{ mg·g}^{-1}\text{·d}^{-1}$). At the waxing stage, the urease activity was highest in the T2 samples ($0.68 \text{ mg·g}^{-1}\text{·d}^{-1}$, which was a maximum of 22.10% higher than in the other treatment groups). At the maturity stage, the T4 samples had the highest urease activity, which was 12.78%, 2.99%, and 7.82% higher than in the T1, T2, and T3 samples, respectively.

In the soil collected from Longfu, the overall urease activity in the T3 samples remained high across the developmental stages. The urease activity was highest in the T3 samples at the tiller bloom and pregnancy spike stages, then highest in the T4 samples at the tassel and waxing stages. At the maturity stage, the T3 samples again had the highest urease activity ($0.78$ mg·g$^{-1}$·d$^{-1}$); this was 7.20%, 1.13%, and 2.07% higher than in the T1, T2, and T4 samples, respectively.

### 3.4.2. Acid Phosphatase

Consistently with the urease activity, the acid phosphatase activity at the maturity stage was highest in the soil collected from Longfu and lowest in the Hehua samples (Figure 3). In the soil collected from Hehua, the acid phosphatase activity was significantly higher in T4 than in the other samples at the early stages of fertility (the tillering, pregnancy spike, and tasseling stages). At the waxing stage, the T2 samples had 15.42%, 10.83%, and 12.62% higher activity than the T1, T3, and T4 samples, but the differences were not significant. At the maturity stage, the acid phosphatase activity was highest in the T3 samples ($0.56$ mg·g$^{-1}$·h$^{-1}$).

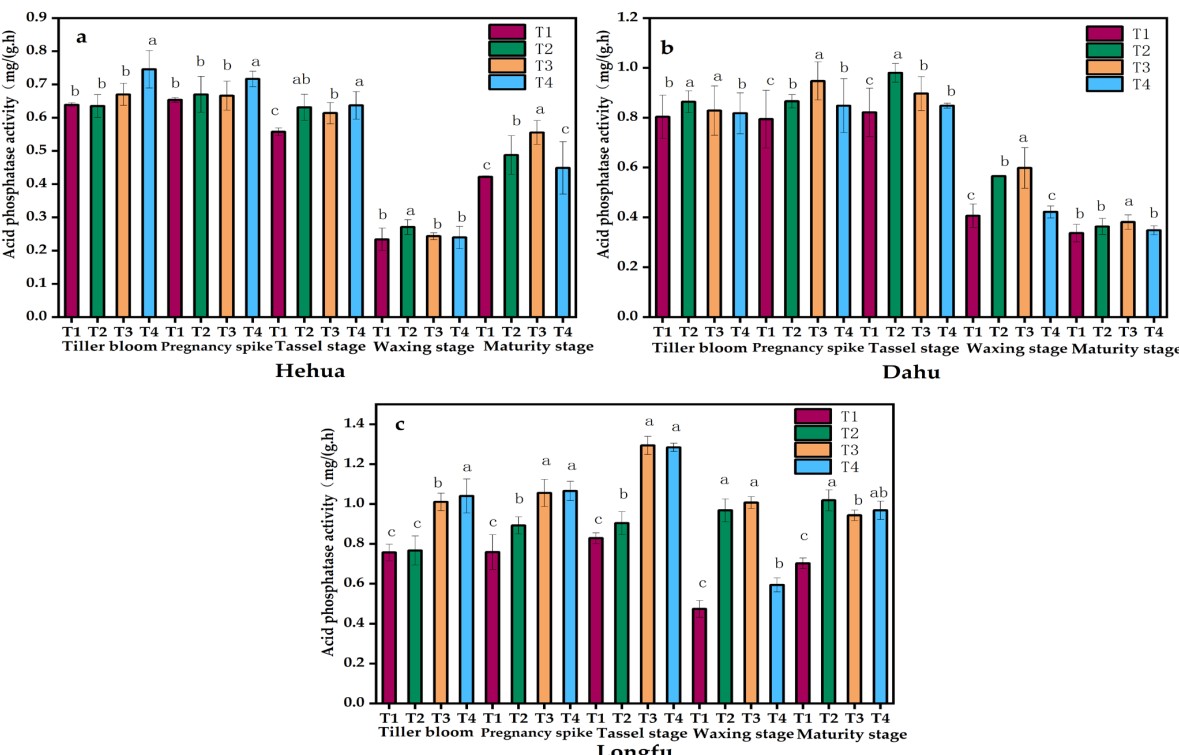

**Figure 3.** At each reproductive stage of rice development (**a**) is acid phosphatase activity in soil samples collected from Hehua; (**b**) is acid phosphatase activity in soil samples collected from Dahu; (**c**) is acid phosphatase activity in soil samples collected from Longfu. T1, no fertilizer; T2, standard fertilizer; T3, 40% organic fertilizer; T4, 100% chemical fertilizer. Lowercase letters above each bar indicate statistically significant groups at $p < 0.05$ (ANOVA).

In the soil collected from Dahu, the acid phosphatase activity remained high in the T2 and T3 samples across the growth stages, but the differences between treatment groups at maturity were not significant. At the tillering stage, the acid phosphatase activity was highest in the T2 group. At the pregnancy spike, waxing, and maturity stages, the T3 samples had the highest activity. In particular, the T3 samples were up to 47.63%, 6.13%, 42.21% higher than the T1, T2, and T4 samples at the waxing stage. At the tasseling stage, the acid phosphatase activity was highest in the T2 samples ($0.98$ mg·g$^{-1}$·h$^{-1}$), which was 19.34%, 9.28%, and 15.53% higher than in the T1, T3, and T4 samples, respectively.

In the soil collected from Longfu, the acid phosphatase activity was significantly higher in the T3 and T4 groups than in the T1 and T2 treatments at the early stages of fertility. For example, the acid phosphatase activity was 37.41% and 40.95% higher in the T4 sample than in the T1 sample at the tiller bloom and pregnancy spike stages, respectively. At the tassel stage, the T3 samples had the highest acid phosphatase activity at 1.29 mg·g$^{-1}$·h$^{-1}$, which was 55.63%, 42.62%, and 0.44% higher than those of the T1, T2, and T4 samples, respectively. At the waxing stage, the acid phosphatase activity was higher in the T3 samples by 112.97%, 4.29%, and 69.92% than in the T1, T2, and T4 samples, respectively. At the maturity stage, the T2 samples had the highest activity at 1.02 mg·g$^{-1}$·h$^{-1}$; this was 45.11%, 8.14%, and 5.32% higher than in the T1, T3, and T4 samples.

### 3.4.3. Sucrase

Overall, the sucrase activity at the maturity stage was highest in Longfu, lower in Dahu, and lowest in Hehua. All three fertilizer application regimens significantly improved the soil sucrase activity compared to the corresponding controls (Figure 4). In the samples collected from Hehua, the sucrase activity remained high in the early stages, then significantly decreased in the later stages. This was likely due to adequate nutrition in the early stages provided by the basal application of fertilizer, followed by nutrient depletion in the later stages due to the poor base soil quality. The sucrase activity generally peaked at the pregnancy spike stage. However, in the T2 treatment group, the sucrase activity was higher at the tiller bloom, tassel, and maturity stages. At the pregnancy spike stage, the sucrase activity was 41.93%, 5.88%, and 24.80% higher in the T4 treatment (30.38 mg·g$^{-1}$·d$^{-1}$) than in the T1, T2, and T4 samples. At the waxing stage, the T3 samples had the highest sucrase activity at 14.31 mg·g$^{-1}$·d$^{-1}$, which was 35.34%, 23.34%, and 25.36% higher than in the T1, T2, and T4 samples, respectively.

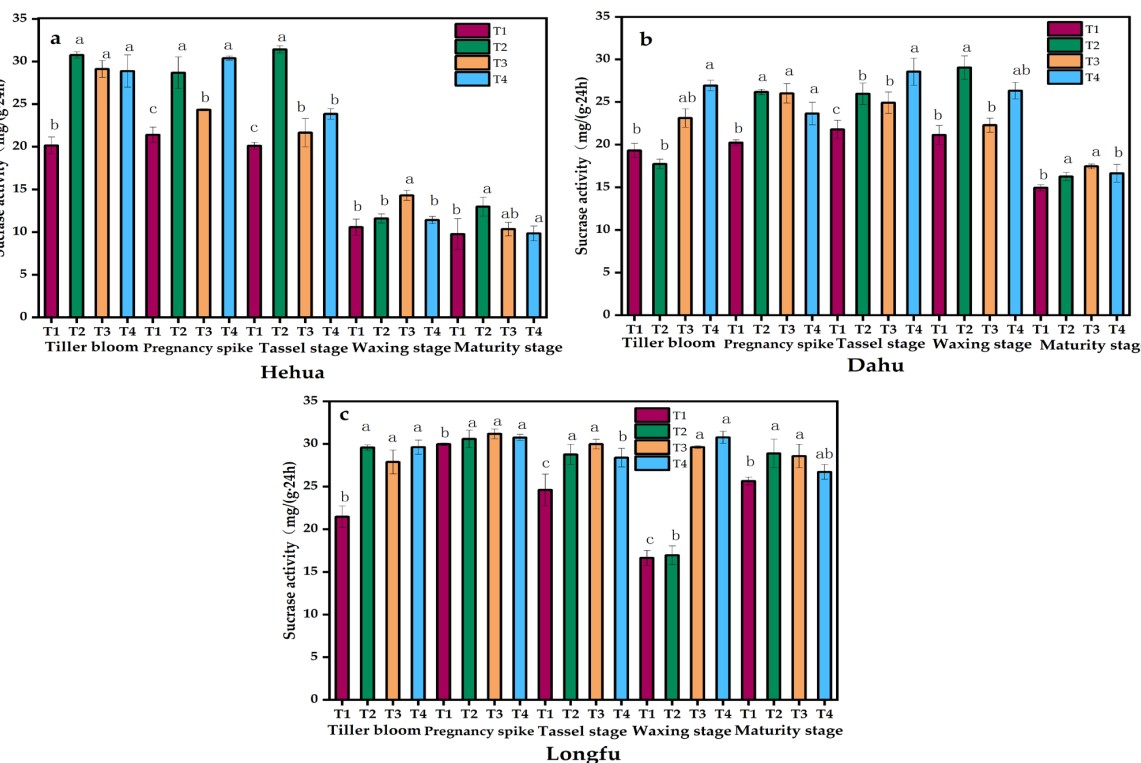

**Figure 4.** At each reproductive stage of rice development (**a**) is sucrase activity in soil samples collected from Hehua; (**b**) is sucrase activity in soil samples collected from Dahu; (**c**) is sucrase activity in soil samples collected from Longfu. T1, no fertilizer; T2, standard fertilizer; T3, 40% organic fertilizer; T4, 100% chemical fertilizer. Lowercase letters above each bar indicate statistically significant groups at *p* < 0.05 (ANOVA).

In soil collected from Dahu, the sucrase activity was highest in the T4 samples at the tillering and milking stages. During the pregnancy spike and waxing stages, the T2 samples had the highest sucrase activity. At the maturity stage, the highest sucrase activity was found in the T3 samples (17.45 mg·g$^{-1}$·d$^{-1}$); this was 16.80%, 7.25%, and 4.87% higher than in the T1, T2, and T4 samples, respectively.

In soil collected from Longfu, there were no significant differences in sucrase activity between treatments in the early growth stages. In the later stages, the differences between the T2, T3, and T4 treatments were significant, although the differences between the T2 and T3 treatments were only significant at the waxing stage. Across all treatment groups, the sucrase activity was highest at the pregnancy spike stage, but the differences between treatments were not significant at that timepoint.

### 3.4.4. Combined Soil Enzyme Activity Index

A combined soil enzyme activity index (GMea) was used to assess the overall key enzyme activity in each soil sample. The GMea was higher in the Longfu and Dahu samples than in the Hehua samples; furthermore, it was higher for all three fertilizer treatment groups compared to the control (Figure 5). The GMea values for the Longfu and Dahu samples were 0.67–0.77 and 0.87–1.15 higher, respectively, than for the Hehua samples. Among the Hehua samples, the GMea was highest in T2, followed by T3 and then T4, and it was lowest in the T1 samples; for Longfu and Dahu, the GMea was higher in T3 than in T2, but it remained the lowest in T1 and second-lowest in T4.

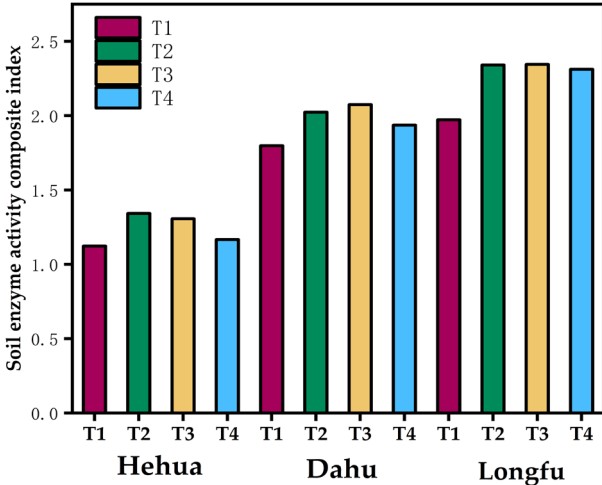

**Figure 5.** Comprehensive enzyme activity index values in soil samples collected from Hehua, Longfu, and Dahu at each reproductive stage of rice development. T1, no fertilizer; T2, standard fertilizer; T3, 40% organic fertilizer; T4, 100% chemical fertilizer.

### 3.4.5. Analysis of Simple Interactions between Location and Treatment with Indicators

In Figure 6, one can see that the T2, T3, and T4 treatments had interactive benefits in Hehua, Dahu, and Longfu.

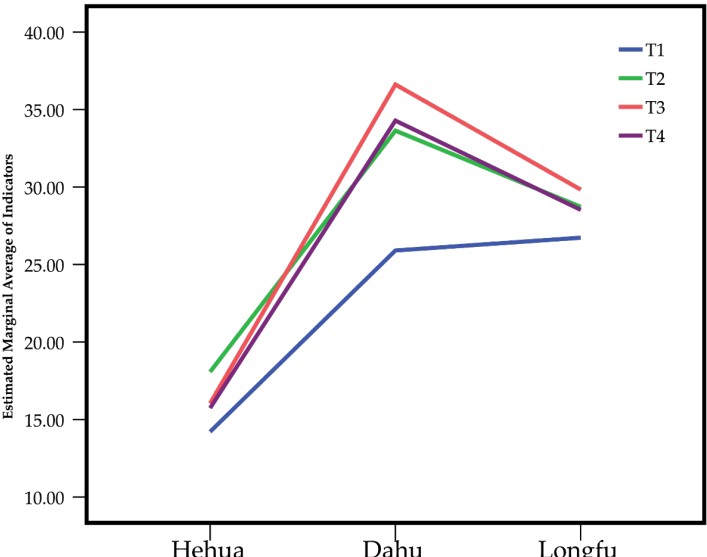

**Figure 6.** Analysis of the mutual benefits among the treatment sites of Hehua, Dahu, and Longfu. T1, no fertilizer; T2, standard fertilizer; T3, 40% organic fertilizer; T4, 100% chemical fertilizer. The indicators were total N, total P, organic matter, alkaline nitrogen at maturity, active phosphorus at maturity, phosphatase at maturity, urease at maturity, and sucrase at maturity.

Analysis of Simple Interactions between Location and Metrics

As can be seen in Figure 7, the relevant indicators had significant differences in the interaction benefits for Hehua, Dahu, and Longfu, with Dahu being the highest and Hehua being the lowest.

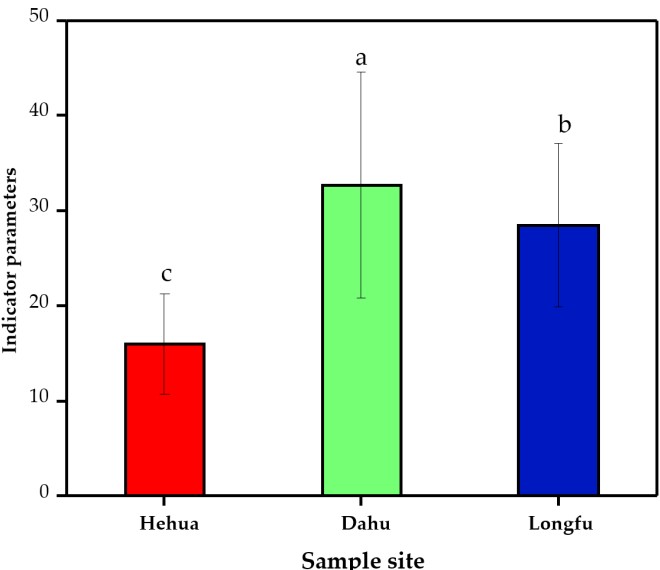

**Figure 7.** Analysis of the interaction benefits of each indicator on Hehua, Dahu, and Longfu. The indicators were total N, total P, organic matter, alkaline nitrogen at maturity, active phosphorus at maturity, phosphatase at maturity, urease at maturity, and sucrase at maturity. Lowercase letters above each bar indicate statistically significant groups at $p < 0.05$ (ANOVA).

Analysis of Simple Interactions between Treatments and Metrics

As can be learned from Figure 8, there were also large differences in the mutual benefits between the treatments and the indicators, with treatments T2 and T3 with organic

fertilizer being significantly higher than treatment T1 without fertilizer and treatment T4 with chemical fertilizer only.

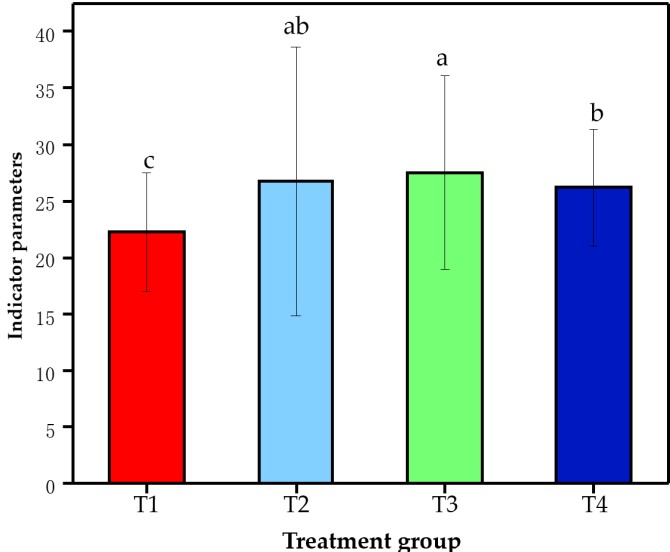

**Figure 8.** Analysis of the benefits of interactions between treatments and metrics. T1, no fertilizer; T2, standard fertilizer; T3, 40% organic fertilizer; T4, 100% chemical fertilizer. The indicators were total N, total P, organic matter, alkaline nitrogen at maturity, active phosphorus at maturity, phosphatase at maturity, urease at maturity, and sucrase at maturity. Lowercase letters above each bar indicate statistically significant groups at $p < 0.05$ (ANOVA).

## 4. Discussion

Long-term application of organic fertilizer significantly increases soil organic matter, total N, and total phosphorus compared with untreated soil [32]. Similarly, organic matter, total N, and available phosphorus levels are increased in soil treated with purely chemical fertilizer treatment or a non-organic fertilizer combination compared to untreated soil [33]. In the present study, we found varying degrees of increases in soil organic matter, total N, and total phosphorus content in three soil collection sites treated with different fertilizer types compared to untreated soil (Table 5). The largest increases in total N and organic matter content occurred at the Hehua site in the sample treated with 100% chemical fertilizer (Table 5). However, all three soil nutrient indicators (total N, available phosphorus, and organic matter content) were also increased in the Hehua and Longfu plots that were treated with 40% organic fertilizer or standard fertilizer, the latter of which included 5% pig manure (Table 5). In low-fertility fields, chemical fertilizers were shown to directly and effectively improve soil fertility. However, the inorganic N contained in chemical fertilizers decomposes quickly and is easily lost, whereas the N contained in organic fertilizers decomposes slowly and is more easily retained in the soil [34]. Therefore, in the long term, combined organic and inorganic fertilization is an effective measure in fields with varying fertility levels.

One of the key parameters assessed in this study was the decomposed alkaline N content in each soil sample. At the maturity stage, in the Hehua samples, the 40% organic (T3), standard (T2), and chemical (T4) fertilizer treatments were associated with 23.24%, 7.01%, and 6.19% increases in decomposed alkaline N compared to untreated soil; in Dahu, the T3, T2, and T4 treatments were associated with decomposed alkaline N increases of 32.00%, 50.32%, and 40.56%, respectively, and in Longfu, the same treatments were associated with increases of 0.62%, 3.57%, and 1.86%, respectively (Table 6). In Hehua, the standard treatment containing 5% pig manure (T2) significantly increased the decomposed alkaline N content compared to the 40% organic fertilizer (T3) treatment. In contrast, the highest alkaline N increases were obtained in Dahu and Longfu soil treated with 40%

organic fertilizer (Table 6). Organic fertilizers, particularly pig manure, reduce organic N mineralization [35,36]. Soil alkaline N content is also closely related to water content and heat conditions. For example, a previous study showed that soil alkaline N content was decreased by 17.84% after flooding [37]. The increases in alkaline N as a result of fertilizer addition were lower in Longfu than in Hehua, which was likely because the Longfu rice field was flooded during the rice maturity phase.

Overall, the organic matter content was lower in the standard fertilizer than in the 40% organic fertilizer treatment, the latter of which included zoysia and rice straw. However, the rate and total amount of organic matter decomposition in the 40% organic treatment was low in the short term. This was in contrast to the high rate of manure decomposition, which facilitates rapid uptake of nutrients by low-fertility rice fields. The slower decomposition of the T3 treatment could result in organic matter accumulation in low-yielding fields (Table 5). In contrast, in medium- and high-fertility fields, the base soil contains a higher abundance of microorganisms with increased species diversity, which promotes the fast decomposition of organic matter. Thus, the application of 40% organic fertilizer would be the most appropriate for fields with higher base fertility levels, and standard fertilizer treatment would be a better option for low-yielding fields.

The available phosphorus content was higher in all fertilized samples than in the control samples, and the organic and standard fertilizer treatments were associated with higher phosphorus content than the 100% chemical fertilizer treatment (Table 6). Generally, fertilizer application increases the total soil phosphorus content, although phosphorus is easily fixed by soil sorption [38]. Compared with chemical fertilizers, organic fertilizers increase levels of highly mobile organic phosphorus, which, in turn, increases the accumulation of active-state phosphorus and, thus, of effective soil phosphorus [39]. This may be because organic fertilizer decomposition produces substances that occupy some of the adsorption sites on the surfaces of iron and aluminum oxides, thus reducing soil adsorption of phosphorus and increasing the amount of active phosphorus in the soil [40]. Organic manure and pig manure mixed with inorganic fertilizer were shown here to improve soil nutrient utilization and enzyme activity, consistently with the findings of Zhao et al. [41].

Soil enzymes are closely related to soil microorganisms and play an important role in catalytic reactions for organic matter decomposition and nutrient cycling [42]. Fertilizer application affects the abundance, species diversity, and metabolic processes of soil microorganisms, which, in turn, alter the levels of soil enzyme activity [43]. Here, the Dahu and Longfu sites were found to have higher urease activity than the Hehua samples did, which may have been due to the base N content of the soils (Figure 2). Across all three sites, urease activity was higher in the soils treated with 40% organic fertilizer or standard fertilizer than in those treated with 100% chemical fertilizer or no fertilizer (Figure 2). This was similar to the results of Deng et al. [44]. On the one hand, exogenous enzymes may be contained in organic fertilizers, which create a good living environment for soil microorganisms and are conducive to the improvement of soil enzyme activities. On the other hand, the application of organic matter increases organic nitrogen, provides abundant energy substances, and enhances the metabolic activities of animals, plants, and microorganisms in the soil, thereby increasing enzyme activities [45–47]. Obviously, the response of the urease activity to organic fertilizers was greater than that to chemical fertilizers, and the soil acid phosphatase activity (Figure 3), sucrase activity (Figure 4), and urease activity (Figure 2) in the three lands were similar.

Soil phosphatase accelerates the rate at which organic phosphorus is dephosphorylated; its activity directly affects the decomposition, conversion, and biological effectiveness of soil organic phosphorus [48]. Long-term application of chemical fertilizers in combination with pig manure may increase soil phosphatase activity [26]. Soil sucrase is associated with carbon cycling in the soil and can be used as a marker of a soil's ability to decompose and utilize organic carbon [49]. Numerous studies have shown that organic and inorganic fertilizer application can effectively increase soil sucrase activity [48]. In addition to directly measuring urease, phosphatase, and sucrase activity, we also assessed soil enzyme activity

by using a comprehensive indicator of soil biological quality, GMea [20]. Overall, GMea was significantly higher in the fertilized samples than in the control samples, as expected. Furthermore, GMea was higher in samples treated with 40% organic fertilizer or 5% pig manure than in those treated with 100% chemical fertilizer (Figure 5). As discussed above, this may have been because organic matter input promoted soil microbe growth and reproduction, increasing the activity of key enzymes in the soil. However, few types of organic fertilizers were tested in the present study; future research should include additional types of organic fertilizers to determine their effects on soil enzymatic activity.

## 5. Conclusions

Experiments testing three types of organic and inorganic fertilizer treatments revealed varying effects on soil parameters in rice fields with different base fertility levels. In Hehua, 100% chemical fertilizer application significantly increased the soil nutrient contents. Plant matter was determined to be a suboptimal fertilizer for low-fertility soils due to the slow rate of decomposition; in such conditions, a relatively high volume of fertilizer should be applied to compensate for the poor soil quality, and the proportion of inorganic fertilizer should be high. In Dahu and Longfu, where organic fertilizer cultivation was more effective due to the higher base soil fertility, the proportion of organic fertilizer could be increased, although the optimal materials and rates require further testing and optimization. Furthermore, the total amount of fertilizer applied to high-fertility fields should be reduced to decrease investment costs, improve fertilizer utilization, and minimize environmental pollution. Overall, the results of this study support the application of organic fertilizer in combination with inorganic fertilizer as an effective measure for the improvement of soil quality in low- and medium-yielding fields. Our findings serve as a valuable guide for rational and economical improvement of soil conditions in fields with a range of yield levels, ultimately promoting increased crop yield and food security.

**Author Contributions:** Conceptualization, G.M. and Y.D.; methodology, S.C.; software, W.T.; validation, G.M., S.C. and W.H.; formal analysis, S.Q. and N.T.; investigation, W.H.; resources G.M.; data curation, Y.D.; writing—original draft preparation, G.M. and S.C.; writing—review and editing, G.M.; visualization, G.M.; supervision, G.M.; project administration, S.Q.; funding acquisition, G.M. All authors have read and agreed to the published version of the manuscript.

**Funding:** Major Science and Technology Program of Hainan Province (No. ZDKJ202001).

**Data Availability Statement:** Data will be made available upon personal request.

**Conflicts of Interest:** The authors declare no conflict of interest.

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
