# Peer review of "Effects of Organic and Inorganic Fertilizers on Soil Nutrient Conditions in Rice Fields with Varying Soil Fertility"

_land, doi:10.3390/land12051026_

Round 1
Reviewer 1 Report
In the introduction there are paragraphs that are not supported by any bibliographical reference. You have to support them with more quotes. For example, between lines 32 and 37. Or lines 39-42. Line 171-173 also needs to be supported by a bibliographical reference.
In the discussion, the citations of the Tables and Figures would be missing to guide the reader.
To conclude, I would add a principal component analysis as well as a correlation analysis. It would be interesting to have a global vision of the study.
Author Response
Reply to the first reviewer's comments
Point 1 In the introduction there are paragraphs that are not supported by any bibliographical reference. You have to support them with more quotes. For example, between lines 32 and 37. Or lines 39-42. Line 171-173 also needs to be supported by a bibliographical reference.
Response 1 Dear reviewers, thank you for your valuable comments. I have added just the right amount of references in the Introduction based on your valuable comments, thank you.
Point 2 In the discussion, the citations of the Tables and Figures would be missing to guide the reader.
Response 2 Dear reviewers, thank you for your valuable suggestions. According to your suggestion, I have added relevant tables and figures in the discussion section to guide readers to read and understand more conveniently.
Point 3 To conclude, I would add a principal component analysis as well as a correlation analysis. It would be interesting to have a global vision of the study.
Response 3 Dear reviewer, thank you for your valuable suggestions. According to your valuable suggestions, I have done relevant principal component analysis and correlation analysis, but while doing principal component analysis, I found that my processing is relatively small (4 processing), There are many indicators, which also leads to the failure of the principal component analysis to display the relevant results of the fourth treatment. If the corresponding indicators are reduced to 3~4, I personally think that it is not very meaningful to do principal component analysis, so I did not continue to do principal component analysis. At the same time, I separated the three regions and made a correlation analysis, which is easier to understand. Thinking about the relevance of the analysis is as follows
Table1 Correlation analysis of each index of Hehua
* indicates significant at the p<0.05 level
Table2 Correlation analysis of each index of Dahu
* and ** indicate significant at the p<0.05 and p<0.01 levels, respectively
Table3 Correlation analysis of each index of Longfu
* and ** indicate significant at the p<0.05 and p<0.01 levels, respectively

Reviewer 2 Report
The study on the effect of organic and inorganic fertilizers on soil nutrient conditions in the rice ecosystem is not new. However, the authors tried well to present the report on how the availability of nutrients is being influenced by nutrient management practices with varying levels of inherent soil fertility.
I have the following observations for the present study
· The abstract may be recast with the inclusion of the effect of the treatments on soil combined enzyme activity.
· Line-14: What are physiological soil indicators?
· Line-21: Fast-acting nutrients mean water soluble? If so better to use water-soluble nutrients throughout the manuscript.
· Line-35&37: What are fertilizer cultivation techniques?
· In the materials and methods section, the year of soil sampling must be mentioned to find out the overall effects of nutrient management practices over the years. Crop management practices need to be elaborately mentioned. Water management in the rice ecosystem is very crucial since the oxidation-reduction potential influence the rate of crop residue decomposition and SOM accumulation and also nutrient availability. Therefore, water management throughout the growth period needs to be mentioned (in simple quantum of irrigation in each growth stage).
· Line-124: Kindly check the different growth stages of rice.
· Mentioned the initial soil bulk density and porosity values to ascertain the effects of nutrient management on these two parameters.
· Effective phosphorus/ fast-acting phosphorus/ available phosphorus- one term may be used throughout the manuscript to avoid confusion.
· Interaction among the sites can be presented in the tables for a better understanding of the influence of inputs on soil properties.
· Figures 2, 3 & 4- look very clumsy. The authors may think up for any other ways of presentation if possible (may be in table or average values across the sites in fig.).
· For line 361-363: Kindly put a suitable reference.
Minor editing is required.
Author Response
Reply to the second reviewer's comments
Point 1 The abstract may be recast with the inclusion of the effect of the treatments on soil combined enzyme activity.
Response 1 Dear reviewer, thank you for your valuable suggestion. I have added the relevant description of the effect of treatment on soil comprehensive enzyme activity according to your suggestion. Thank you. As follows
The effect of organic fertilizer application on the combined enzyme activity of the soil was also higher than that of chemical fertilizers alone.
Point 2 Line-14: What are physiological soil indicators?
Response 2 Dear reviewer, thank you for pointing out that I have changed "soil physiological indicators" to "soil physical indicators" thank you
Point 3 Line-21: Fast-acting nutrients mean water soluble? If so better to use water-soluble nutrients throughout the manuscript.
Response 3 Dear reviewer, the available nutrients in the soil mentioned in the article include the sum of water-soluble nutrients and exchangeable nutrients in the soil.
Point 4 Line-35&37: What are fertilizer cultivation techniques?
Response 4 Dear reviewer, I apologize for my English expression, I have changed to soil improvement technology in the article, the changes are as follows
Soil productivity, fertilizer application techniques, and soil improvement technology vary between areas with different base soil fertility rates Developing standard protocols for reasonable fertilizer application and soil improvement technology based on local soil fertility is therefore of great significance in rice production
Point 5 In the materials and methods section, the year of soil sampling must be mentioned to find out the overall effects of nutrient management practices over the years. Crop management practices need to be elaborately mentioned. Water management in the rice ecosystem is very crucial since the oxidation-reduction potential influence the rate of crop residue decomposition and SOM accumulation and also nutrient availability. Therefore, water management throughout the growth period needs to be mentioned (in simple quantum of irrigation in each growth stage).
Response 5 Dear Reviewer, Thank you for your valuable comments. Based on your comments I have made changes and added sampling time, crop management methods and moisture management. As follows
1 The soil sampling time for this test is 2018.
2 Each experimental plot was constructed with field ridges (20 cm wide and 30 cm high) and wrapped with plastic film to prevent fertilizer and water infiltration between the plots, and each plot was single-rowed and single-irrigated. Other field management practices were consistent with those of the local one-season rice cropping system, in-cluding weed, pest and disease control.
3 The water management of the whole reproductive period is based on shallow water transplanting, inch water rejuvenation, shallow water tillering, sufficient seedlings for sunning, inch water for spike, and wet and strong seeds.
Point 6 Line-124: Kindly check the different growth stages of rice.
Response 6 Dear reviewer, thank you for your review comments. I have made revisions based on the different growth stages of rice you mentioned. Do you think it is reasonable? Thank you. Modify as follows
Soil samples were collected at a depth of 0 to 20 cm at the tiller bloom stages, pregnancy spike stages, tassel stages, waxing stages, and maturity stages.
Point 7 Mentioned the initial soil bulk density and porosity values to ascertain the effects of nutrient management on these two parameters.
Response 7 Dear reviewers, thank you for your valuable comments. I have determined the effect of nutrient management on these two parameters based on your comments, but there are region-specific differences between these parameters. Thank you.
Point 8 Effective phosphorus/ fast-acting phosphorus/ available phosphorus- one term may be used throughout the manuscript to avoid confusion.
Response 8 Dear Reviewer, Thank you for your valuable comments. I have changed all of them according to your comments to -available phosphorus
Point 9 Interaction among the sites can be presented in the tables for a better understanding of the influence of inputs on soil properties.
Response 9 Dear Reviewers, Thank you for your valuable comments. As for your suggestion that the interaction between the sites can be listed in the table, I have done the correlation analysis of the relevant indicators for each of the three sites according to your comments, thank you. As follows
Table1 Correlation analysis of each index of Hehua
* indicates significant at the p<0.05 level
Table2 Correlation analysis of each index of Dahu
* and ** indicate significant at the p<0.05 and p<0.01 levels, respectively
Table3 Correlation analysis of each index of Longfu
* and ** indicate significant at the p<0.05 and p<0.01 levels, respectively
Point 10 Figures 2, 3 & 4- look very clumsy. The authors may think up for any other ways of presentation if possible (may be in table or average values across the sites in fig.).
Response 10 Dear Reviewer, Thank you for your valuable comments. I have made all the adjustments according to your comments.
Point 11 For line 361-363: Kindly put a suitable reference.
Response 11 Dear Reviewers, Thank you for your valuable comments. As for your suggestion that the sentence requires a reference, I have not found any relevant literature so far, so I have deleted it. Thank you.

Reviewer 3 Report
In the study, it is mentioned that very low or moderate fertilization is applied in the majority of agricultural lands and that insufficient fertilization in these areas limits the crop yield. In addition, fertilizer applications increase the productivity of soil quality and the effects of such treatments vary depending on the basic soil fertility. In addition, the implications of different fertilization applications of different fertilizer types are not yet clearly defined. In this study, parameters that act as indicators of chemical and biological soil health in rice fields in red soil areas of different fertilizer types were examined in detail. In general, the study used a fluent language compatible with the journal. However, some edits must be made before the article is accepted.
Specific comments:
Please support these below sentences with properly referenses otherwise this section is lack.
-"Improving soil quality in such fields to increase grain crop yields is an effective method of increasing food security and promoting the strategy of promoting the Chinese strategic national initiative of agricul- tural land and technique development. Soil productivity, fertilizer application tech- niques, and fertilizer cultivation techniques vary between areas with different base soil fertility rates. Developing standard protocols for reasonable fertilizer application and effective fertilizer cultivation based on local soil fertility is therefore of great significance in rice production"
- Please rephrases this sentence " English should be improved. Nitrogen (N) is a key nutrient required for crop growth and development. Low-fertility soils are generally lower in soluble N than high- and medium-fertility soils are.
- There are many unnessarry information from line 57 to 60. Please improve and mention regarding different N fertilizer type.
- what does mean standard fertilizer, please explain this information more clearly.
- I do not know, Even the Author measure many different enzymatic activity, I could not see regarding fertilizer and enzymatic activity interaction in the introduction. So, Please add more information about this shortness.
- These discussion are so general. So, Please focus only the obtained results " Soil microorganisms secrete numerous enzymes that play critical roles in organic 381 matter decomposition and nutrient cycling [32]. Fertilizer application affects the abun- 382 dance, species diversity, and metabolic processes of soil microorganisms, which in turn 383 alter the levels of soil enzyme activity [33]. Here, the Dahu and Longfu sites were found 384 to have higher urease activity than Hehua samples did, which may have been due to the 385 base N content of the soils. Across all three sites, urease activity was higher in the soils 386 treated with 40% organic fertilizer or standard fertilizer than 100% chemical fertilizer or 387 no fertilizer. This was similar to the results of Deng et al. [34], who found that a mixture 388 of inorganic and organic fertilizer significantly increases urease in soils. This may be be- 389 cause application of organic fertilizer increases organic N, which provides abundant en- 390 ergy sources and enhances the metabolic activities of plants and microorganisms living in 391 the soil. Moreover, organic fertilizers may contain exogenous enzymes, enhancing con- 392 ditions for soil microbes and thus further increasing soil enzyme activities [35-37]"
Best Regards
Please recheck some sentences. Because there are some grammatical confirmation.
Author Response
Reply to the third reviewer's comments
Point 1 Please support these below sentences with properly referenses otherwise this section is lack "Improving soil quality in such fields to increase grain crop yields is an effective method of increasing food security and promoting the strategy of promoting the Chinese strategic national initiative of agricul- tural land and technique development. Soil productivity, fertilizer application tech- niques, and fertilizer cultivation techniques vary between areas with different base soil fertility rates. Developing standard protocols for reasonable fertilizer application and effective fertilizer cultivation based on local soil fertility is therefore of great significance in rice production”
Response 1 Dear Reviewer, I have made the relevant changes and added the relevant references based on your valuable suggestions, as follows
Shen Renfang; Wang Chao; Sun Bo. Soil science and technology in the implementation of the strategy of "hiding food in the land and hiding food in technology". Proceedings of the Chinese Academy of Sciences 2018, 135-144.
Point 2 Please rephrases this sentence " English should be improved. Nitrogen (N) is a key nutrient required for crop growth and development. Low-fertility soils are generally lower in soluble N than high- and medium-fertility soils are.
Response 2 Dear Reviewer I have made changes based on your suggestions, do you think it makes sense? "Nitrogen (N) is a key nutrient required for crop growth and development. Soluble N is generally higher in high and medium fertility soils than in low fertility soils".
Point 3 There are many unnecessary information from line 57 to 60. Please improve and mention regarding different N fertilizer type.
Response 3 Dear reviewers I have made changes based on your valuable suggestions. Thank you. Nitrogen (N) is a key nutrient required for crop growth and development. Soluble N is generally higher in high and medium fertility soils than in low fertility soils. Tao et al. [1] showed that high-yield have relatively high levels of organic matter and alkaline nitrogen in the soil. Increase soil organic matter to promote soil total nitrogen and alkaline nitrogen fixation
Point 4 what does mean standard fertilizer, please explain this information more clearly.
Response 4 Dear reviewer, thank you for your valuable opinion. standard fertilization means that the fertilization standard is based on the fertilization habits of farmers in the test site.
Point 5 I do not know, Even the Author measure many different enzymatic activity, I could not see regarding fertilizer and enzymatic activity interaction in the introduction. So, Please add more information about this shortness.
Response 5 Dear reviewers, thank you for your valuable comments. Based on your valuable comments, the author has supplemented relevant expositions. Do you think it is reasonable? Thanks
Ye Xie Feng et al. showed that tilling green manure significantly increased the enzymatic activity and fertility level of soil, and the highest enzymatic activity and fertility level was achieved when the tilling volume was 22,500-30,000 kg/hm2. In low fertility soils, the number and activity of microorganisms in the soil sink increased significantly with organic-inorganic application, which had a significant effect on improving soil fertility.Liu et alshowed that the long-term combined application of chemical fertilizers and pig manure could improve phosphatase activity in the soil, and the application of organic fertilizers could The organic fertilizer application can improve the soil structure and fertility. Similarly, when the straw application rate was 11250 kg/hm2, the number of fungi, bacteria, actinomycetes and the activities of invertase and cellulase in the soil increased the most, and when the straw application rate was 7500 kg/hm2, soil alkaline urease, phosphatase activity and alkaline nitrogen content increased significantly.
Point 6 These discussion are so general. So, Please focus only the obtained results " Soil microorganisms secrete numerous enzymes that play critical roles in organic matter decomposition and nutrient cycling [32]. Fertilizer application affects the abundance, species diversity, and metabolic processes of soil microorganisms, which in turn alter the levels of soil enzyme activity [33]. Here, the Dahu and Longfu sites were found to have higher urease activity than Hehua samples did, which may have been due to the base N content of the soils. Across all three sites, urease activity was higher in the soils treated with 40% organic fertilizer or standard fertilizer than 100% chemical fertilizer or no fertilizer. This was similar to the results of Deng et al. [34], who found that a mixture of inorganic and organic fertilizer significantly increases urease in soils. This may be be cause application of organic fertilizer increases organic N, which provides abundant energy sources and enhances the metabolic activities of plants and microorganisms living in the soil. Moreover, organic fertilizers may contain exogenous enzymes, enhancing conditions for soil microbes and thus further increasing soil enzyme activities [35-37]"
Response 6- Dear reviewers, thank you for your valuable comments. Based on your valuable comments, the author expresses the discussion here in detail. Do you think it is reasonable? Thanks,modify as follows
Soil enzymes are closely related to soil microorganisms and play an important role in catalytic reactions for organic matter decomposition and nutrient cycling. Fertilizer application affects the abundance, species diversity, and metabolic processes of soil microorganisms, which in turn alter the levels of soil enzyme activity. Here, the Dahu and Longfu sites were found to have higher urease activity than Hehua samples did, which may have been due to the base N content of the soils. Across all three sites, urease activity was higher in the soils treated with 40% organic fertilizer or standard fertilizer than 100% chemical fertilizer or no fertilizer. This was similar to the results of Deng et al. On the one hand, exogenous enzymes may be contained in organic fertilizers, which create a good living environment for soil microorganisms and are conducive to the improvement of soil enzyme activities. On the other hand, the application of organic matter increases organic nitrogen, provides abundant energy substances, and enhances the metabolic activities of soil animals, plants and microorganisms, thereby increasing enzyme activities. Obviously, the response of urease activity to organic fertilizers was greater than that of chemical fertilizers, and the soil acid phosphatase activity, sucrase activity and urease activity in the three lands were similar.

Round 2
Reviewer 2 Report
The manuscript has been improved after incorporating the comments and suggestions. Still, I have some clarifications on the manuscript as follows.
Line 141-142: Growth stages of rice. Whether tassel stage occurs in rice or maize?
Line 347: Lotus?
Interaction study is not simply the correlations between the parameters. Interaction studies for sites and treatments are done to know whether the effects of treatments are consistent in different sites or not. After the table, the authors can put the ANOVA table and state whether parameters are significant in w.r.t to sites, treatments and their interaction. (If possible the authors can present the tables like this). The correlation studies presented in table 7. 8 & 9 may not be required.
ANOVA
Site
Treatment
Site x Treatment
Good wishes
Author Response
Reply to the 2nd reviewer's comments
Point 1 Line 141-142: Growth stages of rice. Whether tassel stage occurs in rice or maize?
Response 1 Dear Reviewer Thank you for your valuable comments, based on your comments I have made the following changes
Soil samples were collected at a depth of 0 to 20 cm at the tiller bloom stages, pregnancy spike stages, tassel stages, waxing stages, and maturity stages of rice.
Point 2 Line 347: Lotus?
Response 2 Dear Reviewer Thank you for your valuable comments, I am very sorry, I have deleted them, thank you.
Point 3 Interaction study is not simply the correlations between the parameters. Interaction studies for sites and treatments are done to know whether the effects of treatments are consistent in different sites or not. After the table, the authors can put the ANOVA table and state whether parameters are significant in w.r.t to sites, treatments and their interaction. (If possible the authors can present the tables like this). The correlation studies presented in table 7. 8 & 9 may not be required.
Response 3 Dear Reviewer Thank you for your valuable comments, according to your comments I have deleted the correlation between indicators in the article, and combined your comments to do the interaction between three regions and four treatments; each indicator has done the analysis of the interaction between location and treatment respectively, do you think it is reasonable? Thank you for your valuable comments. as follows;
1 Analysis of simple interactions between location and treatment by indicators
Figure 6 we can see that the T2,T3,T4 treatments have interactive benefits with Hehua, Dahu, and Longfu.
Fig. 6 Analysis of mutual benefits among Hehua, Dahu, Longfu and the treatment sites. T1, no fertilizer; T2, standard fertilizer; T3, 40% organic fertilizer; T4, 100% chemical fertilizer. Indicators (total N, total p, organic matter, alkaline nitrogen at maturity, active phosphorus at maturity, phosphatase at maturity, urease at maturity, sucrase at maturity)
2 Simple interaction analysis between location and metrics
As can be seen from Figure 7, the relevant indicators had significant differences in the interaction benefits for Hehua, Dahu and Longfu, with Dahu being the highest andHehua the lowest.
Fig. 7 Analysis of the interaction benefits of each indicator on Hehua, Dahu, and Longfu. Indicators (total N, total p, organic matter, alkaline nitrogen at maturity, active phosphorus at maturity, phosphatase at maturity, urease at maturity, sucrase at maturity) Lowercase letters above each bar indicate statistical significance groups at p < 0.05 (ANOVA).
3 Simple interaction analysis between treatment and metrics
As can be learned from Figure 8, there were also large differences in the mutual benefits between the treatments and the indicators, with treatments T2 and T3 with organic fertilizer being significantly higher than treatments T1 without fertilizer and T4 with chemical fertilizer only.
Fig. 8 Interaction benefit analysis between treatment and metrics. T1, no fertilizer; T2, standard fertilizer; T3, 40% organic fertilizer; T4, 100% chemical fertilizer. Indicators (total N, total p, organic matter, alkaline nitrogen at maturity, active phosphorus at maturity, phosphatase at maturity, urease at maturity, sucrase at maturity). Lowercase letters above each bar indicate statistical significance groups at p < 0.05 (ANOVA).

Reviewer 3 Report
Author improved paper regarding reviewer suggestions. Now, the paper could be accepted for the publication.
Best Regards
Author Response
Response to the second commenter's comment
Dear Reviewer,
Thank you very much for your approval of this paper and for the comments and suggestions you gave for this paper, they were very helpful to me, thank you again for your hard work on this paper, thank you!
Best wishes,
Guozhu Ma